# Monitoring of Serum Potassium and Calcium Levels in End-Stage Renal Disease Patients by ECG Depolarization Morphology Analysis

**DOI:** 10.3390/s22082951

**Published:** 2022-04-12

**Authors:** Hassaan A. Bukhari, Carlos Sánchez, José Esteban Ruiz, Mark Potse, Pablo Laguna, Esther Pueyo

**Affiliations:** 1BSICoS Group, I3A Institute, IIS Aragón, University of Zaragoza, 50018 Zaragoza, Spain; cstapia@unizar.es (C.S.); laguna@unizar.es (P.L.); epueyo@unizar.es (E.P.); 2CIBER en Bioingeniería, Biomateriales y Nanomedicina (CIBER-BBN), 50018 Zaragoza, Spain; 3Carmen Team, Inria Bordeaux—Sud-Ouest, 33405 Talence, France; mark@potse.nl; 4Université de Bordeaux, IMB, UMR 5251, 33400 Talence, France; 5Nephrology Department, Hospital Clínico Universitario Lozano Blesa, 50009 Zaragoza, Spain; jeruizl@salud.aragon.es

**Keywords:** electrocardiogram, QRS complex morphology, T wave morphology, potassium estimation, calcium estimation, hemodialysis, time warping, regression

## Abstract

Objective: Non-invasive estimation of serum potassium, $\left[K^{+}\right]$, and calcium, $\left[{\mathrm{Ca}}^{2+}\right]$, can help to prevent life-threatening ventricular arrhythmias in patients with advanced renal disease, but current methods for estimation of electrolyte levels have limitations. We aimed to develop new markers based on the morphology of the QRS complex of the electrocardiogram (ECG). Methods: ECG recordings from 29 patients undergoing hemodialysis (HD) were processed. Mean warped QRS complexes were computed in two-minute windows at the start of an HD session, at the end of each HD hour and 48 h after it. We quantified QRS width, amplitude and the proposed QRS morphology-based markers that were computed by warping techniques. Reference $\left[K^{+}\right]$ and $\left[{\mathrm{Ca}}^{2+}\right]$ were determined from blood samples acquired at the time points where the markers were estimated. Linear regression models were used to estimate electrolyte levels from the QRS markers individually and in combination with T wave morphology markers. Leave-one-out cross-validation was used to assess the performance of the estimators. Results: All markers, except for QRS width, strongly correlated with $\left[K^{+}\right]$ (median Pearson correlation coefficients, *r*, ranging from 0.81 to 0.87) and with $\left[{\mathrm{Ca}}^{2+}\right]$ (*r* ranging from 0.61 to 0.76). QRS morphology markers showed very low sensitivity to heart rate (HR). Actual and estimated serum electrolyte levels differed, on average, by less than 0.035 mM (relative error of 0.018) for $\left[K^{+}\right]$ and 0.010 mM (relative error of 0.004) for $\left[{\mathrm{Ca}}^{2+}\right]$ when patient-specific multivariable estimators combining QRS and T wave markers were used. Conclusion: QRS morphological markers allow non-invasive estimation of $\left[K^{+}\right]$ and $\left[{\mathrm{Ca}}^{2+}\right]$ with low sensitivity to HR. The estimation performance is improved when multivariable models, including T wave markers, are considered. Significance: Markers based on the QRS complex of the ECG could contribute to non-invasive monitoring of serum electrolyte levels and arrhythmia risk prediction in patients with renal disease.

## 1. Introduction

Patients with chronic kidney disease (CKD), particularly at the most severe stages, are at high risk for life-threatening arrhythmias and sudden cardiac death [1,2,3,4], and millions of them die every year due to lack of access to affordable treatment. Between five and seven million end-stage renal disease (ESRD) patients need renal replacement therapy worldwide [5]. ESRD patients undergoing hemodialysis (HD) frequently have serum electrolyte levels outside normal ranges, which can increase the mortality risk [1,2,3,4].

Non-invasive monitoring of electrolyte levels can be useful for risk prediction and the triggering of early warnings. Electrocardiogram (ECG) ventricular depolarization and repolarization are, in particular, altered by variations in serum potassium ($\left[K^{+}\right]$) and calcium ($\left[{\mathrm{Ca}}^{2+}\right]$) [1,2,6,7,8]. ECG markers have been proposed for the continuous monitoring of these two electrolyte levels, which could facilitate timely therapies for ESRD patients.

Most ECG-based estimators of $\left[K^{+}\right]$ and $\left[{\mathrm{Ca}}^{2+}\right]$, such as those based on the QT interval or on T wave amplitude, slope and slope-to-amplitude ratio, are based exclusively on ventricular repolarization and commonly refer to a specific time interval, to an amplitude at a given time point or to a particular portion of the T wave [9,10,11,12,13,14,15]. Although the QT interval has long been used to monitor CKD patients during HD [16,17,18,19,20,21], contradictory findings have been presented, with many studies reporting QT prolongation [18,19,21,22,23] and others showing QT shortening or no effects during HD [24,25]. Other recent studies have investigated changes in markers of sympathetic activity-related T wave instability during HD [26] and in model-based descriptors of T wave morphology in the interdialytic interval [27]. These studies, however, have not established a tight correlation between the proposed markers and $\left[K^{+}\right]$, thus limiting their possibilities for ambulatory $\left[K^{+}\right]$ monitoring. In previous studies, we used nonlinear dynamics and time-warping techniques to characterize changes in the whole T wave at varying $\left[K^{+}\right]$ and $\left[{\mathrm{Ca}}^{2+}\right]$ in patients and simulated ECGs [28,29,30,31,32,33]. We found a strong relationship between $\left[K^{+}\right]$ and T wave linear and nonlinear features.

Assessment of ventricular depolarization for serum electrolyte estimation has been less explored. Similarly to QT studies, research on QRS complex duration has rendered inconsistent results, with some works reporting widened QRS complexes [1,34] and others reporting narrowed QRS complexes at high $\left[K^{+}\right]$ [2,35]. Other works have assessed the time voltage area, amplitude and sine wave shape of the QRS complex, but limitations in terms of their significance or their dependence on blood volume have been acknowledged [14,36,37,38]. None of these studies have, however, quantitatively characterized overall variations in the morphology of the QRS complex during HD. We hypothesized that QRS morphology could provide complementary information on $\left[K^{+}\right]$ and $\left[{\mathrm{Ca}}^{2+}\right]$, in addition to that provided by repolarization.

The aim of this study was to quantify changes in the QRS amplitude, duration and morphology, the latter both in the time and amplitude domains, at varying $\left[K^{+}\right]$, $\left[{\mathrm{Ca}}^{2+}\right]$ and heart rate (HR) in ESRD patients. Univariable and multivariable electrolyte estimators, including novel QRS morphological markers in combination with already proposed T wave markers [28,29,30,39], were devised, and the contribution of depolarization analysis to $\left[K^{+}\right]$ and $\left[{\mathrm{Ca}}^{2+}\right]$ monitoring was assessed.

## 2. Materials

Forty-eight-hour 12-lead ECGs with 3.75 μV resolution and a sampling frequency of 1 kHz (H12+, Mortara Instruments, Milwaukee, WI, USA) were acquired from 29 ESRD patients of Hospital Clínico Universitario de Zaragoza (HCUZ). ECG acquisition started 5 minutes before HD and lasted for 48 hours (Figure 1, bottom blue line), with patients in the supine position. Concurrently, six blood samples were taken during and after the HD session: the first one at HD onset and the next three samples every hour during the HD session (Figure 1, $h_{0}$ to $h_{3}$ in red). The fifth blood sample ($h_{4}$) was obtained at the end of the HD (minute 215 or 245, depending on the patient) while the sixth blood sample was taken after 48 h ($h_{48}$), immediately before the next HD session. Serum $\left[K^{+}\right]$ and $\left[{\mathrm{Ca}}^{2+}\right]$ were measured at those six time points, as shown in Figure 1 [28,29,30], using a Cobas 6000 c501 analyzer (Roche Diagnostics, Germany) by an indirect ion selective electrode method. We defined another time point for ECG analysis, corresponding to a segment taken two minutes before the end of HD (minute 213 or 243, depending on the patient), which we denoted by $h_{4}^{-}$. The $\left[K^{+}\right]$ value at $h_{4}^{-}$ was assumed to be the same as at $h_{4}$, as the time difference between these two segments was just two minutes. All patients signed informed consent. The study protocol was approved by the Research Ethics Committee of Aragón (*CEICA*, ref. PI18/003, 25 January 2018). Table 1 shows the population characteristics.

## 3. Methods

A flow chart showing all the stages of the signal processing starting with 12-lead ECG pre-processing, followed by principal component analysis (PCA) transformation, computation of mean warped QRS (MWQRS) complexes and QRS morphological markers, and finishing with the estimation of $\left[K^{+}\right]$ and $\left[{\mathrm{Ca}}^{2+}\right]$ is shown in Figure 2.

### 3.1. ECG Pre-Processing

A band-pass filter (0.5 to 40 Hz) was used to remove baseline wander, muscular noise and powerline interference from measured ECG signals. A wavelet-based single-lead delineation method [40] was used for QRS detection and wave delineation. Spatial principal component (PC) analysis was performed on the complete ECG of the eight independent leads [41] to enhance QRS complex energy. PCs were obtained from the eigenvectors of the 8×8 inter-lead auto-correlation matrix of QRS complexes extracted from a stable (in terms of HR) 10-min ECG segment at the end of the HD session to emphasize the QRS complex components. This segment corresponded to the time when the patient was discharged from hospital with restored serum $\left[K^{+}\right]$ and $\left[{\mathrm{Ca}}^{2+}\right]$. The ECG recording from each time point during and after HD was subsequently projected onto the direction of the first PC. The QRS complexes from each time point in the first PC were delineated [40] to mark their onsets, peaks and ends.

### 3.2. QRS Descriptors

#### 3.2.1. Computation of Mean Warped QRS Complexes

MWQRS complexes, which are optimal representative averages in both temporal and amplitude domains, were calculated from the two-minute ECG segment at the end of each HD hour following the methodology described in [39]. The two-minute ECG segment was short enough to maintain the assumption of stability for both electrolyte and HR values [30]. Only QRS complexes in each analyzed two-minute window that presented the dominant polarity were considered for MWQRS calculation, with the polarity defined as:(1)$$\begin{array}{cc} p_{0} & =\left\{\begin{array}{cc} 1, & \mathrm{if}\max_{n}\left\{|f\left(n\right)|\right\}=\max_{n}\left\{f\left(n\right)\right\}\\ -1, & \mathrm{if}\max_{n}\left\{|f\left(n\right)|\right\}=-\min_{n}\left\{f\left(n\right)\right\}, \end{array}\right. \end{array}$$
where f(n) represents the QRS complex under analysis. An average of 92% of QRS complexes in each analyzed two-minute ECG segment were found to present such dominant polarity (see Appendix A). An initial MWQRS was computed after aligning the QRS complexes having the dominant polarity with respect to their gravity center so that the calculated MWQRS was not affected by potential outlier QRS complex polarities and morphologies [39]. Outliers, defined as those QRS complexes having a Spearman’s correlation coefficient with an initial MWQRS lower than 0.98, were rejected. The final MWQRS was computed from the remaining QRS complexes.

The QRS descriptors defined in the next sections were computed from MWQRS complexes for ECG segments during and after HD ($h_{0}$, $h_{1}$, $h_{2}$, $h_{3}$, $h_{4}^{-}$, $h_{4}$ and $h_{48}$).

#### 3.2.2. QRS Duration and Amplitude Markers

For each MWQRS, the following descriptors were computed: $QRS_{w}$, which represented the QRS width calculated from QRS onset to end (expressed in ms) [40].$QRS_{a}$, which represented the QRS amplitude calculated from the minimum to maximum amplitude of the QRS complex (expressed in mV).

#### 3.2.3. QRS Morphology Markers

Morphology-based QRS descriptors were computed using the time-warping methodology, as previously described [39]. For each patient, a reference QRS complex, $f^{r}\left(t^{r}\right)$, was calculated from the MWQRS at the end of the HD session, the time when the patient was discharged from hospital with restored serum ion levels. Representative QRS complexes for each HD stage, $f^{s}\left(t^{s}\right)$, were calculated from the MWQRS at two-minute windows at the end of each hour during the HD session and 48 h after it. $f^{s}\left(t^{s}\right)$ is expressed as $f^{s}\left(t^{s}\right)={\left[f^{s}\left(t^{s}\left(1\right)\right),...,f^{s}\left(t^{s}\left(N_{s}\right)\right)\right]}^{⊤}$ and the reference QRS complex as $f^{r}\left(t^{r}\right)={\left[f^{r}\left(t^{r}\left(1\right)\right),...,f^{r}\left(t^{r}\left(N_{r}\right)\right)\right]}^{⊤}$. The vectors $t^{r}={\left[t^{r}\left(1\right),...,t^{r}\left(N_{r}\right)\right]}^{⊤}$ and $t^{s}={\left[t^{s}\left(1\right),...,t^{s}\left(N_{s}\right)\right]}^{⊤}$ are the uniformly sampled time vectors corresponding to the QRS complexes $f^{s}$ and $f^{r}$, respectively. $N_{r}$ and $N_{s}$ represent the total duration, in samples, of $t^{r}$ and $t^{s}$, respectively. Figure 3a shows an example of $f^{r}$ and $f^{s}$, with their respective time domains, $t^{r}$ and $t^{s}$.

Let $\gamma (t^{r})$ be the warping function that relates $t^{r}$ and $t^{s}$ such that $f^{s}\left(\gamma \left(t^{r}\right)\right)$ denotes the time-domain warping of $f^{s}\left(t^{s}\right)$ using $\gamma (t^{r})$. The square-root slope function (SRSF) transformation was used to find the optimal warping function [39]. This transformation was defined as:(2)$$q_{f}\left(t\right)=sign\left(\dot{f}\left(t\right)\right)\;{\left|\dot{f}\left(t\right)\right|}^{\frac{1}{2}}.$$

The optimal warping function was determined as the one minimizing the amplitude difference between the SRSF of $f^{r}\left(t^{r}\right)$ and $f^{s}\left(\gamma \left(t^{r}\right)\right)$:(3)$$\gamma^{*}\left(t^{r}\right)=\underset{\gamma \left(t^{r}\right)}{\arg\min}\left(∥q_{f^{r}}\left(t^{r}\right)-q_{f^{s}}\left(\gamma \left(t^{r}\right)\right)\sqrt{\dot{\gamma }\left(t^{r}\right)}∥\right).$$

A dynamic programming algorithm was used to obtain the function $\gamma^{*}\left(t^{r}\right)$ that optimally warped $f^{r}\left(t^{r}\right)$ into $f^{s}\left(t^{s}\right)$. This function is shown in Figure 3d. The warped QRS complex, $f^{s}\left(\gamma^{*}\left(t^{r}\right)\right)$, is shown in Figure 3b, together with the reference QRS complex, $f^{r}\left(t^{r}\right)$.

The descriptor, $d_{w,Q}^{u}$, was used to quantify the level of warping required to optimally align the QRS complexes $f^{s}\left(t^{s}\right)$ and $f^{r}\left(t^{r}\right)$: (4)$$d_{w,Q}^{u}=\frac{1}{N_{r}}\sum_{n=1}^{N_{r}}\left|\gamma^{*}\left(t^{r}\left(n\right)\right)-t^{r}\left(n\right)\right|.$$

The amplitude descriptor, $d_{a,Q}$, was computed from the area contained between $f^{r}\left(t^{r}\right)$ and $f^{s}\left(\gamma^{*}\left(t^{r}\right)\right)$ normalized by the L2-norm of $f^{r}\left(t^{r}\right)$, thus quantifying amplitude differences after time warping the two QRS complexes: (5)$$d_{a,Q}=p_{0}\frac{s_{a}}{|s_{a}|}\frac{∥f^{s}\left(\gamma^{*}\left(t^{r}\right)\right)-f^{r}\left(t^{r}\right)∥}{∥f^{r}\left(t^{r}\right)∥}\times 100,$$
where $s_{a}=\sum_{n=1}^{N_{r}}\left(f^{s}\left(\gamma^{*}\left(t^{r}\left(n\right)\right)\right)-f^{r}\left(t^{r}\left(n\right)\right)\right)$ was used to account for the sign.

The above described marker $d_{w,Q}^{u}$ accounted for both linear and nonlinear warping required to fit the two QRS complexes in the time domain. The nonlinear component marker $d_{w,Q}^{\mathrm{NL}}$ was quantified as: (6)$$d_{w,Q}^{\mathrm{NL}}=\frac{1}{N_{r}}\sum_{n=1}^{N_{r}}\left|\gamma^{*}\left(t^{r}\left(n\right)\right)-\gamma_{l}^{*}\left(t^{r}\left(n\right)\right)\right|,$$
where $\gamma_{l}^{*}\left(t^{r}\right)$ (green line in Figure 3d) was derived by linearly fitting $\gamma^{*}\left(t^{r}\right)$ using the least absolute residual method.

The nonlinear amplitude marker $d_{a,Q}^{\mathrm{NL}}$ was defined by computing the $L_{2}$ norm of the difference between $L_{2}$-normalized versions of $f^{r}\left(t^{r}\right)$ and $f^{s}\left(\gamma^{*}\left(t^{r}\right)\right)$: (7)$$d_{a,Q}^{\mathrm{NL}}=∥\frac{f^{r}\left(t^{r}\right)}{∥f^{r}\left(t^{r}\right)∥}-\frac{f^{s}\left(\gamma^{*}\left(t^{r}\right)\right)}{∥f^{s}\left(\gamma^{*}\left(t^{r}\right)\right)∥}∥\times 100.$$

Figure 3a shows $f^{r}$ and $f^{s}$, referring to their respective time domains, $t^{r}$ and $t^{s}$. The warped QRS complex (red), $f^{s}\left(\gamma^{*}\left(t^{r}\right)\right)$, is shown in Figure 3b, together with the reference QRS complex (blue), $f^{r}\left(t^{r}\right)$.

The analyzed warping-based QRS descriptors, computed from MWQRS during and after HD stages, $h_{i}$, included:$d_{w,Q}^{u}$, which represented temporal variations in QRS morphology (expressed in ms),$d_{a,Q}$, which represented amplitude variations in QRS morphology (expressed as a percentage),$d_{w,Q}^{\mathrm{NL}}$, which represented nonlinear temporal variations in QRS morphology (expressed in ms),$d_{a,Q}^{\mathrm{NL}}$, which represented nonlinear amplitude variations in QRS morphology (expressed as a percentage).

### 3.3. Statistical Analysis

Pearson correlation coefficients (*r*) were computed to assess the strength of the linear relationship between $\left[K^{+}\right]$, $\left[{\mathrm{Ca}}^{2+}\right]$ or RR and each of the investigated QRS descriptors in each patient individually.

A Wilcoxon signed-rank test was applied to test for significant differences (p<0.05) in $\left[K^{+}\right]$, $\left[{\mathrm{Ca}}^{2+}\right]$, RR, $QRS_{w}$, $QRS_{a}$, $d_{w,Q}^{u}$, $d_{a,Q}$, $d_{w,Q}^{\mathrm{NL}}$ and $d_{a,Q}^{\mathrm{NL}}$ between consecutive HD stages. The reason for using a non-parametric statistical test was that, according to the Shapiro–Wilk test, the data distributions were not normal.

To test whether the Pearson correlation coefficient between each QRS marker and $\left[K^{+}\right]$, $\left[{\mathrm{Ca}}^{2+}\right]$ or RR was significantly different from 0, a Student’s *t*-test was used after transforming the statistical distribution of *r* into a normal distribution using Fisher’s z transform [42].

All statistical analyses were performed using MATLAB version R2020b for Windows (MathWorks Inc., Novi, MI, USA).

### 3.4. Uni- and Multivariable Estimation of $\left[K^{+}\right]$ and $\left[{\mathrm{Ca}}^{2+}\right]$

To estimate $\left[K^{+}\right]$ and $\left[{\mathrm{Ca}}^{2+}\right]$, univariable and multivariable estimators were developed. Of all the analyzed QRS descriptors, $d_{w,Q}^{u}$ was selected to build univariable estimators because it presented high median absolute Pearson correlation with $\left[K^{+}\right]$ and $\left[{\mathrm{Ca}}^{2+}\right]$ and a relatively low IQR range, particularly for $\left[K^{+}\right]$. Univariable estimators were additionally built using the repolarization descriptor $d_{w,T}^{u}$, which had shown strong correlation with electrolyte levels [29,30]. $d_{w,T}^{u}$ was calculated analogously to $d_{w,Q}^{u}$ but for the T wave instead of the QRS complex. Multivariable estimators were tested using both $d_{w,Q}^{u}$ and $d_{w,T}^{u}$.

The univariable $\left[{\hat{K}}^{+}\right]{}_{u}$ ($\left[\hat{C}a^{2+}\right]{}_{u}$, respectively) and multivariable $\left[{\hat{K}}^{+}\right]{}_{m}$ ($\left[\hat{C}a^{2+}\right]{}_{m}$, respectively) estimators were of the following form: (8)$$I_{d_{w,Q}^{u}}=\beta_{0}^{d_{w,Q}^{u}}+\beta_{1}^{d_{w,Q}^{u}}\cdot d_{w,Q}^{u}\;,$$
(9)$$I_{d_{w,T}^{u}}=\beta_{0}^{d_{w,T}^{u}}+\beta_{1}^{d_{w,T}^{u}}\cdot d_{w,T}^{u}\;,$$
and
(10)$$I_{m}=\beta_{0}^{m}+\beta_{1}^{m}\cdot d_{w,Q}^{u}\;+\beta_{2}^{m}\cdot d_{w,T}^{u},$$
where $I\in \{\left[{\hat{K}}^{+}\right]\;,\left[\hat{C}a^{2+}\right]\;\}$. The performance of the estimators was tested by using the leave-one-out cross-validation approach.

For the univariable estimators, the coefficient vector $\beta ={\left[\begin{array}{cc} \beta_{0}^{d_{w,Q}^{u}} & \beta_{1}^{d_{w,Q}^{u}} \end{array}\right]}^{T}$ or $\beta ={\left[\begin{array}{cc} \beta_{0}^{d_{w,T}^{u}} & \beta_{1}^{d_{w,T}^{u}} \end{array}\right]}^{T}$ was estimated as [43,44]: (11)$$\hat{\beta }={\left(X^{T}X\right)}^{-1}X^{T}y^{T},$$
with $X=\left[\begin{array}{cc} j^{T} & x_{b}^{T} \end{array}\right]$. Three different types of estimators were considered, namely stage-specific, patient-specific and global estimators, as described in the following paragraphs. The definition of $j^{T}$, $x_{b}^{T}$ and y was different for each type of estimator:For an HD stage-specific (S) estimator, which estimates the electrolyte level at stage $h_{i}$ of a given patient *q* from the marker’s values of the remaining patients at that stage, the vector $\hat{\beta }$ was calculated from the vector $j=\left[\begin{array}{cccc} 1 & 1 & \cdots & 1 \end{array}\right]$ of dimension 1×Q, with *Q* being the total number of patients minus 1, the vector $x_{b}=\left[\begin{array}{cccc} b_{i,1} & b_{i,2} & \cdots & b_{i,Q} \end{array}\right]$ containing the values of the marker *b* = $d_{w,Q}^{u}$ or $d_{w,T}^{u}$ at stage *i* from patients 1, …, *Q* (all except for patient *q*) and the vector $y=\left[\begin{array}{cccc} \left[K^{+}\right]{}_{i,1} & \left[K^{+}\right]{}_{i,2} & \cdots & \left[K^{+}\right]{}_{i,Q} \end{array}\right]$ containing the measured $\left[K^{+}\right]$ values at stage *i*. The vector y was defined analogously for $\left[{\mathrm{Ca}}^{2+}\right]$. This procedure was carried out for each HD stage, $h_{i}$, separately.For a patient-specific (P) estimator, which estimates the electrolyte level at stage $h_{i}$ of a given patient *q* from the marker’s values at the remaining stages for that same patient, the vector $\hat{\beta }$ was calculated from the vector $j=\left[\begin{array}{ccccc} 1 & 1 & \cdots & 1 & 1 \end{array}\right]$ of dimension 1 × 6 (if $h_{i}$ is $h_{0}$ or $h_{48}$) or 1 × 7 (if $h_{i}$ was different from $h_{0}$ or $h_{48}$), $x_{b}$ = $\left[\begin{array}{cccccc} b_{0,q} & b_{0,q} & b_{1,q} & \cdots b_{4^{-},q} & b_{48,q} & b_{48,q} \end{array}\right]$ containing the values of the marker *b* = $d_{w,Q}^{u}$ or $d_{w,T}^{u}$ for patient *q* at the different time points except for $h_{i}$, with $h_{0}$ and $h_{48}$ being duplicated. The vector y was defined as $y\;=\;[\;\left[K^{+}\right]{}_{0,q}\;\left[K^{+}\right]{}_{0,q}\;\left[K^{+}\right]{}_{1,q}\;\cdots \left[K^{+}\right]{}_{4^{-},q}\;\left[K^{+}\right]{}_{48,q}\;\left[K^{+}\right]{}_{48,q}\;]$ containing the measured $\left[K^{+}\right]$ values for patient *q* at all time points except for $h_{i}$. An analogous definition of vector y was applied for $\left[{\mathrm{Ca}}^{2+}\right]$. This process was repeated for each patient individually.For a global (G) estimator, which estimates the electrolyte level at stage $h_{i}$ of a given patient *q* from the marker values at all other time points from all other patients, the vector $\hat{\beta }$ was calculated by defining vectors $x_{b}$ and y to contain the marker values and the electrolyte measures from all patients at all stages except for patient *q* at time *i*.

For the multivariable estimators, $\beta ={\left[\begin{array}{ccc} \beta_{0}^{m} & \beta_{1}^{m} & \beta_{2}^{m} \end{array}\right]}^{T}$ was estimated using Equation (Equation 11), where the matrix $X=\left[\begin{array}{ccc} j^{T} & x_{b^{\left(1\right)}}^{T} & x_{b^{\left(2\right)}}^{T} \end{array}\right]$, with $b^{\left(1\right)}$ = $d_{w,Q}^{u}$ and $b^{\left(2\right)}$ = $d_{w,T}^{u}$, was calculated as described for the univariable estimators depending on the type of estimator (stage-specific, patient-specific or global).

The error, *e*, between actual $\left[K^{+}\right]$ (or $\left[{\mathrm{Ca}}^{2+}\right]$) and estimated $\left[{\hat{K}}^{+}\right]$ (or $\left[\hat{C}a^{2+}\right]$) was computed as
(12)$$e=\left[K^{+}\right]-\left[{\hat{K}}^{+}\right].$$

The relative error, $e_{r}$, with respect to the range of electrolyte measurements was computed as
(13)$$e_{r}=\frac{\left[K^{+}\right]-\left[{\hat{K}}^{+}\right]}{\left[K^{+}\right]{}_{R}}.$$
where $\left[K^{+}\right]{}_{R}$ was defined as the difference between the 75th and 25th percentiles of $\left[K^{+}\right]$ across patients at each HD stage. An analogous definition for $\left[{\mathrm{Ca}}^{2+}\right]$ was used.

The relative error, $e_{v}$, with respect to the actual electrolyte measurement $\left[K^{+}\right]$ for each patient at each HD stage (analogously for $\left[{\mathrm{Ca}}^{2+}\right]$) was computed as
(14)$$e_{v}=\frac{\left[K^{+}\right]-\left[{\hat{K}}^{+}\right]}{\left[K^{+}\right]}.$$

The mean absolute error was computed by averaging the absolute value of *e* defined in Equation (Equation 12). The root mean square error was computed by taking the root mean square of *e* defined in Equation (Equation 12).

To assess the agreement between actual and estimated $\left[K^{+}\right]$ and $\left[{\mathrm{Ca}}^{2+}\right]$ values, Bland–Altman analysis was performed [45], in which the differences between the actual and estimated $\left[K^{+}\right]$ and $\left[{\mathrm{Ca}}^{2+}\right]$ were plotted as a function of their averages, for all patients at all HD time points (see Appendix A). $\left[K^{+}\right]$ and $\left[{\mathrm{Ca}}^{2+}\right]$ estimation accuracy using our investigated markers was also compared with those of the previously proposed markers $T_{S/A}$ [10,11] and $T_{S/\sqrt{A}}$ [12] (see Appendix A):$T_{S/A}$ represented the ratio between the maximal downward slope (in absolute value) and the amplitude of the T wave [10,11].$T_{S/\sqrt{A}}$ represented the ratio between the maximal downward slope (in absolute value) and the square root of the amplitude of the T wave [12].

It should be noted that estimations of $\left[K^{+}\right]$ and $\left[{\mathrm{Ca}}^{2+}\right]$ could not be performed at the end of the HD session ($h_{4}$) since the morphological QRS complex markers were zero by definition due to the fact that the reference was taken at that time stage. Therefore, we defined an extra time point $h_{4}^{-}$ just before the HD end (reference) so as to improve the estimation accuracy based on one additional HD point.

## 4. Results

### 4.1. Characterization of QRS Complex Changes during and after HD

The top panels in Figure 4 show the results for the QRS markers ($QRS_{w}$, $QRS_{a}$, $d_{w,Q}^{u}$, $d_{a,Q}$, $d_{w,Q}^{\mathrm{NL}}$ and $d_{a,Q}^{\mathrm{NL}}$) together with $\left[K^{+}\right]$, $\left[{\mathrm{Ca}}^{2+}\right]$ and RR variations during and after HD for all patients. Statistically significant differences between consecutive HD stages were found for $QRS_{a}$, $d_{w,Q}^{u}$, $d_{a,Q}$, $d_{w,Q}^{\mathrm{NL}}$ and $d_{a,Q}^{\mathrm{NL}}$ as well as for $\left[K^{+}\right]$ and $\left[{\mathrm{Ca}}^{2+}\right]$ variations, but not for RR.

The bottom panels of Figure 4 show MWQRS during and after HD (red) compared to the reference MWQRS (blue) computed at the end of HD. Lower $QRS_{a}$ values could be observed at $h_{0}$ and $h_{48}$, corresponding to the highest $\left[K^{+}\right]$ and the lowest $\left[{\mathrm{Ca}}^{2+}\right]$.

### 4.2. Contribution of $\left[K^{+}\right]$, $\left[{\mathrm{Ca}}^{2+}\right]$ and HR Variations to QRS Complex Changes

Figure 5 shows the Pearson correlation coefficient, *r*, between the analyzed QRS markers and $\left[K^{+}\right]$ (black), $\left[{\mathrm{Ca}}^{2+}\right]$ (red) and RR (blue). $QRS_{a}$, $d_{w,Q}^{u}$, $d_{a,Q}$, $d_{w,Q}^{\mathrm{NL}}$ and $d_{a,Q}^{\mathrm{NL}}$ were the most highly correlated, in median, with $\left[K^{+}\right]$ (median *r* being −0.87, 0.78, −0.80, 0.73 and 0.81, respectively) and $\left[{\mathrm{Ca}}^{2+}\right]$ (0.76, −0.61, 0.63, −0.70 and −0.75, respectively). The IQR of *r* was the lowest for $d_{w,Q}^{u}$ when the correlation with $\left[K^{+}\right]$ was analyzed, and for $QRS_{a}$ in the case of $\left[{\mathrm{Ca}}^{2+}\right]$. Poor association between all the analyzed markers and RR was found.

Table 2 shows the *p*-values from the Student’s *t*-test used to determine the statistical significance of non-zero mean Fisher z-transformed Pearson correlation coefficients between QRS markers and each of $\left[K^{+}\right]$, $\left[{\mathrm{Ca}}^{2+}\right]$ and RR in the patient population, during and after HD. All the analyzed QRS markers showed significant association with $\left[K^{+}\right]$ and most of them (all but $QRS_{w}$) with $\left[{\mathrm{Ca}}^{2+}\right]$. On the other hand, no marker presented significant association with RR.

### 4.3. Uni- and Multivariable Estimation of $\left[K^{+}\right]$ and $\left[{\mathrm{Ca}}^{2+}\right]$

Figure 6 shows an illustrative example of the comparison between measured and estimated $\left[K^{+}\right]$ and $\left[{\mathrm{Ca}}^{2+}\right]$ using stage-specific (S), patient-specific (P) and global (G) approaches during ($h_{0}$, $h_{1}$, $h_{2}$, $h_{3}$, $h_{4}^{-}$) and after ($h_{48}$) HD for a particular patient, for both univariable (panels a–b) and multivariable (panel c) estimators. The patient-specific approach provided better results in terms of reduced errors, particularly using multivariable estimation (mean $e_{v}$ over HD points being 0.071 (S), −0.008 (P), 0.001 (G) for $\left[K^{+}\right]$ and 0.019 (S), 0.001 (P), 0.031 (G) for $\left[{\mathrm{Ca}}^{2+}\right]$).

Figure 7 shows the relative errors, $e_{v}$, for all patients and HD stages in the estimation of $\left[K^{+}\right]$ and $\left[{\mathrm{Ca}}^{2+}\right]$ using stage-specific (top panel), patient-specific (middle panel) and global (bottom panel) approaches. Appendix A shows the relative errors, $e_{r}$.

Table 3 shows actual and estimated $\left[K^{+}\right]$ and $\left[{\mathrm{Ca}}^{2+}\right]$ values over the study population at each HD stage. Multivariable estimation results using stage-specific, patient-specific and global approaches are presented.

Table 4 and Table 5 show the median and IQR values of intra-patient Pearson correlation coefficients between actual and estimated $\left[K^{+}\right]$ ($\left[{\mathrm{Ca}}^{2+}\right]$, respectively) using univariable and multivariable estimators.

Bland–Altman plots between actual and estimated $\left[K^{+}\right]$ and $\left[{\mathrm{Ca}}^{2+}\right]$ for the proposed markers are shown in Appendix A).

Table 6 and Table 7 show a comparison between estimation errors obtained for the markers analyzed in this study and in previous studies. Appendix A show mean absolute and root mean square errors for the analyzed markers.

## 5. Discussion

We investigated changes in QRS duration, amplitude and morphology at varying $\left[K^{+}\right]$, $\left[{\mathrm{Ca}}^{2+}\right]$ and HR in ESRD patients. We designed $\left[K^{+}\right]$ and $\left[{\mathrm{Ca}}^{2+}\right]$ estimators based on our proposed QRS morphological characteristics, taken both individually and in combination with T wave morphology markers. We showed the accuracy of our proposed estimators using three different approaches, stage-specific, patient-specific and global estimation, which outperformed previously proposed methods. Our results offered new non-invasive tools to monitor serum $\left[K^{+}\right]$ and $\left[{\mathrm{Ca}}^{2+}\right]$, which could have a significant role in clinical practice and could contribute to reduce the mortality risk associated with abnormal electrolyte levels in ESRD patients.

### 5.1. Characterization of QRS Complex Amplitude, Duration and Morphology in ESRD Patients during and after HD

In ECG recordings of ESRD patients, we evaluated QRS duration and amplitude, measured by markers $QRS_{w}$ and $QRS_{a}$, and QRS morphological characteristics, measured by markers $d_{w,Q}^{u}$, $d_{a,Q}$, $d_{w,Q}^{\mathrm{NL}}$ and $d_{a,Q}^{\mathrm{NL}}$, which was proposed here for the first time. All markers except for $QRS_{w}$ presented significant changes during and after HD. These changes were strongly associated with variations in $\left[K^{+}\right]$ and $\left[{\mathrm{Ca}}^{2+}\right]$ but not in HR. The inconsistent relationship between QRS markers, such as $QRS_{w}$, and HR has been investigated in previous works, including the study by Hnatkova et al. [46], who reported increases in QRS duration with increasing HR in 35% of their patients and decreases in QRS duration in the remaining 65%. In line with these results, we found that QRS became markedly wider at higher RR intervals for 34% of the patients; it became markedly narrower for 21% of the patients; its width moderately changed with RR for 21% of the patients; and it showed poor association with RR for the remaining 24% of the patients (see Appendix A). This led to a median *r* value of 0.42 between $QRS_{w}$ and RR over the 29 ESRD patients, reflecting a notably weaker relationship between $QRS_{w}$ and RR as compared to other depolarization markers. Regarding QRS amplitude, even if we found $QRS_{a}$ to remarkably change during HD, this marker depended on ECG amplitudes at specific time points, which, in noisy ambulatory recordings, could lead to large changes not associated with variations in electrolyte levels. On the other hand, changes in warping-based markers accounted for deviations in the whole QRS waveform and could thus be better suited for ambulatory monitoring. In particular, if QRS duration on top of amplitude changes occurred in the inter-dialytic interval, as, e.g., reported during advanced ischemia [47,48], these changes could be reflected in our proposed QRS warping markers.

On top of investigating the relationship between QRS markers and $\left[K^{+}\right]$ or $\left[{\mathrm{Ca}}^{2+}\right]$, we also investigated the relationship between these markers and sodium concentration ($\left[{\mathrm{Na}}^{+}\right]$). $\left[{\mathrm{Na}}^{+}\right]$ variations were less remarkable than those of $\left[K^{+}\right]$ and $\left[{\mathrm{Ca}}^{2+}\right]$ during HD in our patients (see Appendix A). Importantly, none of the markers were strongly associated with $\left[{\mathrm{Na}}^{+}\right]$ (see Appendix A). $\left[{\mathrm{Na}}^{+}\right]$ could have been expected to play a more important role in modulating depolarization markers because the fast sodium current was primarily responsible for phase 0 of the action potential and its changes might have manifested in QRS complex alterations. However, we found that the QRS complex became markedly wider at higher $\left[{\mathrm{Na}}^{+}\right]$ for 43% of the patients; it became markedly narrower for 7% of the patients; its width moderately changed with $\left[{\mathrm{Na}}^{+}\right]$ for 14% of the patients; and it showed poor association with $\left[{\mathrm{Na}}^{+}\right]$ for the remaining 36% of the patients (see Appendix A).

High inter-individual variability was found in the relationship between the analyzed QRS markers and $\left[K^{+}\right]$ or $\left[{\mathrm{Ca}}^{2+}\right]$. This was especially remarkable in the case of $d_{a,Q}$, which presented high IQR in the intra-patient correlation coefficients with $\left[K^{+}\right]$ and $\left[{\mathrm{Ca}}^{2+}\right]$. Such high dispersion could be explained by QRS polarity effects, as a reduction (increase, respectively) in the absolute amplitude could be reflected as either positive or negative $d_{a,Q}$, depending on QRS being predominantly positive or negative.

Changes in ECG characteristics induced by variations in $\left[K^{+}\right]$ and $\left[{\mathrm{Ca}}^{2+}\right]$ have been extensively investigated in terms of ventricular repolarization. A number of studies have characterized changes in T wave width, amplitude, slope or slope-to-amplituderatio [10,11,13,49,50]. We have recently quantified changes in T wave nonlinear dynamics and morphology and have shown their relationship with $\left[K^{+}\right]$ and $\left[{\mathrm{Ca}}^{2+}\right]$ variations [28,29,31,33,51].

The analysis of electrolyte-induced alterations in ventricular depolarization remains, however, much more limited. In a study including 923 patients with severe hyperkalemia, sine wave-shaped QRS complexes were observed in almost 36.7% of patients [38]. In the present study, we could observe such behavior in most of the patients (Figure 4). Inconsistent results have been reported in relation to the effects of $\left[K^{+}\right]$ on QRS duration, with a larger proportion of studies reporting QRS widening [1,34,52,53,54,55,56] and others reporting QRS narrowing [2,35] with increased $\left[K^{+}\right]$. Here, we observed no significant changes in QRS width during and after HD. Regarding QRS amplitude, we found $QRS_{a}$ to be strongly negatively correlated with $\left[K^{+}\right]$ and positively correlated with $\left[{\mathrm{Ca}}^{2+}\right]$, in accordance with the increase in $QRS_{a}$ with decreasing $\left[K^{+}\right]$, as described by Astan et al. [35]. We observed similar results for the QRS morphology-based amplitude marker $d_{a,Q}$. Our study characterized additional QRS morphological changes by the warping markers $d_{w,Q}^{u}$, $d_{w,Q}^{\mathrm{NL}}$ and $d_{a,Q}^{\mathrm{NL}}$ further extended these results to provide a robust characterization of QRS changes during and after HD in ESRD patients. Figure 8 illustrates QRS complexes and T waves at the start and end of HD.

### 5.2. Multivariable Predictors of $\left[K^{+}\right]$ and $\left[{\mathrm{Ca}}^{2+}\right]$ Based on Depolarization and Repolarization Characteristics

Based on the novel QRS morphology markers of this study and on already proposed T wave markers [29,30,39], we designed linear univariable and multivariable $\left[K^{+}\right]$ and $\left[{\mathrm{Ca}}^{2+}\right]$ estimators. In particular, we used $d_{w,Q}^{u}$ (QRS marker) and $d_{w,T}^{u}$ (T wave marker), as these were strongly correlated with $\left[K^{+}\right]$ and $\left[{\mathrm{Ca}}^{2+}\right]$ and poorly related to each other (see Appendix A), thus potentially providing complementary information to monitor electrolyte variations.

For each of the constructed $\left[K^{+}\right]$ and $\left[{\mathrm{Ca}}^{2+}\right]$ linear estimators, we used stage-specific, patient-specific and global estimation approaches. Overall, the stage-specific approach rendered results with both mean and median estimation errors very close to zero but with higher dispersion than in the patient-specific approach. The latter approach would be suitable for clinical application, as electrolyte levels could be predicted in each patient based on a short ECG recording of the same patient.

Multivariable estimators combining information from $d_{w,Q}^{u}$ and $d_{w,T}^{u}$ outperformed univariable estimators (the ones proposed by us and other authors [10,11,12]). The estimation errors for the multivariable estimators were lower (Figure 7, Appendix A, Table 6 and Table 7) and the correlation coefficients *r* between actual and estimated electrolyte levels were higher than for univariable estimators, both for the patient-specific and global estimation approaches (see Table 4 and Table 5). In particular, for the patient-specific approach, median, *r*, in $\left[K^{+}\right]$ estimation was 0.75 for the combination of $d_{w,Q}^{u}$ and $d_{w,T}^{u}$, while it was 0.56 for $d_{w,Q}^{u}$ and 0.55 for $d_{w,T}^{u}$. Similarly, for the global approach, median, *r*, in $\left[{\mathrm{Ca}}^{2+}\right]$ estimation was 0.70 for the combination compared to 0.64 for $d_{w,Q}^{u}$ and 0.64 for $d_{w,T}^{u}$. It should be noted that these correlation coefficient values were computed between actual $\left[K^{+}\right]$ and estimated $\left[{\hat{K}}^{+}\right]$ while the correlation between each of the tested markers and $\left[K^{+}\right]$ was higher, both for the QRS marker $d_{w,Q}^{u}$ proposed here and for the T wave marker $d_{w,T}^{u}$ analyzed in our previous studies [29,30].

These results supported the use of our proposed QRS markers to improve prediction of $\left[K^{+}\right]$ and $\left[{\mathrm{Ca}}^{2+}\right]$ by ECG repolarization markers. ECG depolarization-based estimators have been scarcely investigated in the literature for serum electrolyte monitoring. In [57], an ECG-based $\left[K^{+}\right]$ estimator was designed using QRS duration in addition to T wave markers, but QRS duration was found not to be highly correlated with $\left[K^{+}\right]$, in agreement with our present results for $QRS_{w}$. Pilia et al. [14] reviewed studies evaluating QRS amplitude and width features, but no improved serum electrolyte prediction by incorporating these features into repolarization-based estimators was provided. Here, we proposed univariable and multivariable $\left[K^{+}\right]$ and $\left[{\mathrm{Ca}}^{2+}\right]$ estimators that included information from the whole morphology of the QRS complex. By accounting for characteristics beyond QRS amplitude and width, these estimators could offer more robust performance for ambulatory monitoring of ESRD patients and overcome some limitations of previously proposed markers, such as their dependence on blood volume [36,37].

For all the estimators we built, we found that the values of $\left[K^{+}\right]$ and $\left[{\mathrm{Ca}}^{2+}\right]$ at the beginning of each of the two HD sessions, i.e., time points $h_{0}$ and $h_{48}$, were the more challenging to estimate, as could be observed from the largest estimation errors at those time points (Figure 4). Here, we duplicated the values of the ECG markers at $h_{0}$ and $h_{48}$ to give them more weight in the training step, as all other measures corresponding to $h_{1}$, $h_{2}$, $h_{3}$ and $h_{4}^{-}$ were more similar to one another and the learning could otherwise be biased towards such measures.

### 5.3. Study Limitations and Future Research

This study investigated 29 ECG recordings of ESRD patients. Although the dataset was originally planned to include a larger number of patients, ECG acquisition had to be stopped due to the situation generated by the COVID-19 pandemic. Future studies should investigate the application of the proposed methods to larger numbers of patients.

For each patient, six blood samples were available, five of them taken during an HD session and the sixth one at the beginning of the following HD session. Future studies could be designed to have more frequent $\left[K^{+}\right]$ and $\left[{\mathrm{Ca}}^{2+}\right]$ measurements, especially during the first HD hour, when electrolyte levels vary most remarkably. This could help to improve the learning of the estimators.

We used linear estimators due to the small number of samples, particularly when using a patient-specific approach. Future work could investigate the use of nonlinear estimators [11,32], which could prove to be particularly relevant for the estimation of $\left[K^{+}\right]$ and $\left[{\mathrm{Ca}}^{2+}\right]$ at the start of HD sessions when these take values far from those at other HD stages.

We did not have access to measurements of blood volume or of other variables that could be used to infer them. Studies on other datasets where such measurements were available could test the relationship between the markers $d_{w,Q}^{u}$ and $d_{w,T}^{u}$ used in our $\left[K^{+}\right]$ and $\left[{\mathrm{Ca}}^{2+}\right]$ estimators and the blood volume. In addition, some of the patients analyzed in the study had diseases, such as diabetes mellitus. We did not find significant differences in the analyzed markers between diabetic and non-diabetic patients. Nevertheless, future studies addressing larger patient cohorts could investigate the impact of diseases additional to ESRD on the relationship between QRS markers and electrolytes.

Some publications have reported the use of deep learning methods for hypo- and hyperkalemia screening from the ECG. While such methods require large amounts of training data and could not be addressed with the database analyzed here, further investigations could extend the present work to include machine learning techniques for the same purpose [58,59,60].

We focused our research on $\left[K^{+}\right]$ and $\left[{\mathrm{Ca}}^{2+}\right]$ estimation. $\left[{\mathrm{Na}}^{+}\right]$ was found to present less notable variations during HD and none of our analyzed QRS markers showed significant association with it in the reduced dataset analyzed in this study, which should be further tested in larger patient cohorts. Although variations in other electrolytes, such as magnesium $\left[{\mathrm{Mg}}^{2+}\right]$, have also been shown to alter the ECG to some extent [2,24,61,62,63], $\left[{\mathrm{Mg}}^{2+}\right]$ measurements were not available for the present study.

An additional future research line related to this work will be aimed at extending the present investigations to include patient-specific electrophysiological simulations in biventricular models embedded in torso models. From those simulations, realistic ECGs could be computed and used to gain understanding on the relationship between ECG characteristics and $\left[K^{+}\right]$ or $\left[{\mathrm{Ca}}^{2+}\right]$.

## 6. Conclusions

Our proposed QRS morphology markers presented remarkable changes during and after HD, which were strongly associated with $\left[K^{+}\right]$ and $\left[{\mathrm{Ca}}^{2+}\right]$ in the ESRD patients. Multivariable estimators based on combined QRS and T wave morphological variability allowed accurate prediction of $\left[K^{+}\right]$ and $\left[{\mathrm{Ca}}^{2+}\right]$, outperforming estimators based on only ECG depolarization or repolarization. These results could pave the way to ambulatory, non-invasive monitoring of electrolyte levels, which could help to prevent fatal ventricular arrhythmias in ESRD patients.

## Figures and Tables

**Figure 1 sensors-22-02951-f001:**
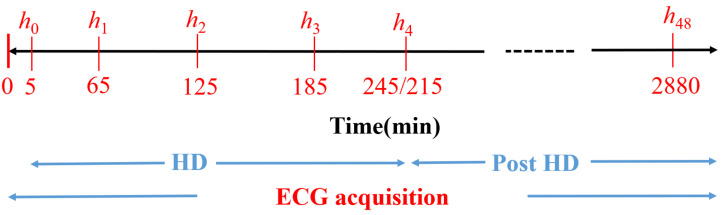
Diagram of the study protocol. $h_{0}$ to $h_{48}$ are the time points (in minutes) for blood sample extraction. Reproduced with permission from Bukhari et al., Computers in Biology and Medicine; published by Elsevier, 2022.

**Figure 2 sensors-22-02951-f002:**
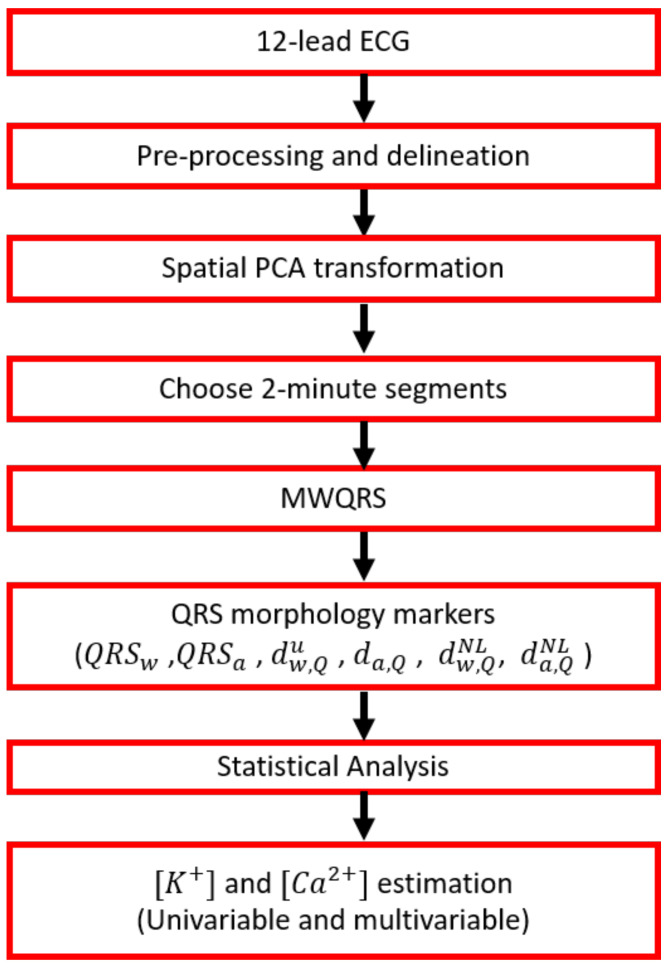
Flow chart showing the ECG processing steps performed in this study, from the collection of raw ECGs to the estimation of $\left[K^{+}\right]$ and $\left[{\mathrm{Ca}}^{2+}\right]$.

**Figure 3 sensors-22-02951-f003:**
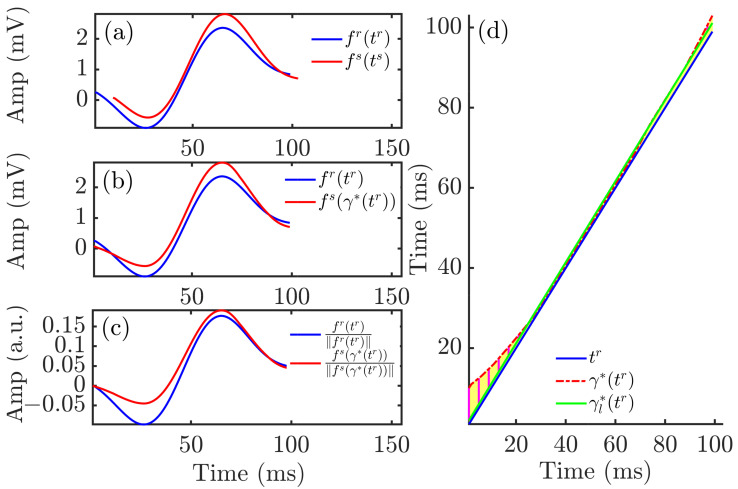
Time warping of QRS complexes. Panel (**a**) shows the reference (blue) and investigated (red) QRS complexes obtained from an ECG segment during HD. Panel (**b**) shows the warped QRS complexes, which had the same duration whilst keeping the original amplitude. Panel (**c**) depicts the warped QRS complexes after normalization by their L2-norms. The yellow area in panel (**d**) represents $d_{w,Q}^{u}$, which quantified the total amount of warping. The green solid line is the linear regression function $\gamma_{l}^{*}\left(t^{r}\right)$ best fitted to $\gamma^{*}\left(t^{r}\right)$.

**Figure 4 sensors-22-02951-f004:**
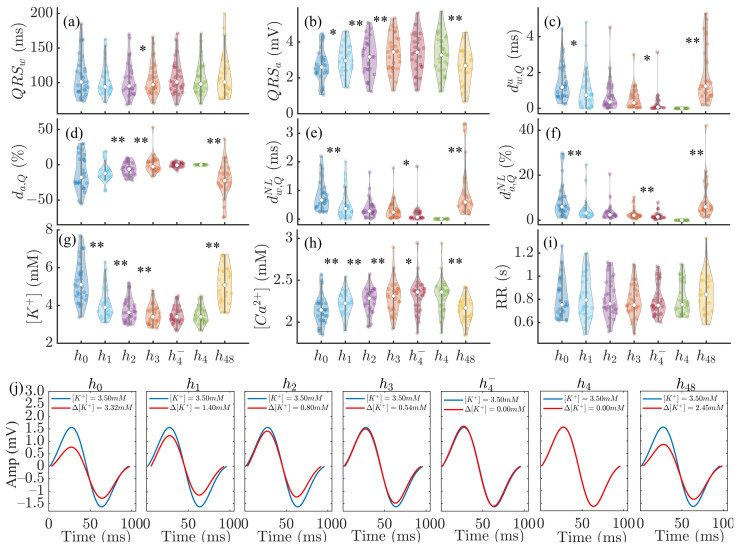
Panels (**a**–**f**): changes in $QRS_{w}$, $QRS_{a}$, $d_{w,Q}^{u}$, $d_{a,Q}$, $d_{w,Q}^{\mathrm{NL}}$ and $d_{a,Q}^{\mathrm{NL}}$ during HD stages. Panels (**g**–**i**): corresponding variations in $\left[K^{+}\right]$, $\left[{\mathrm{Ca}}^{2+}\right]$ and RR. In panels (**a**–**i**), * denotes p<0.05 and ** denotes p<0.01. In each panel, the central white dot indicates the median. Each dot corresponds to an individual patient. Panel (**j**): MWQRS (red) of a patient at different HD stages and reference MWQRS (blue). Δ denotes the change in $\left[K^{+}\right]$ with respect to the end of HD ($h_{4}$).

**Figure 5 sensors-22-02951-f005:**
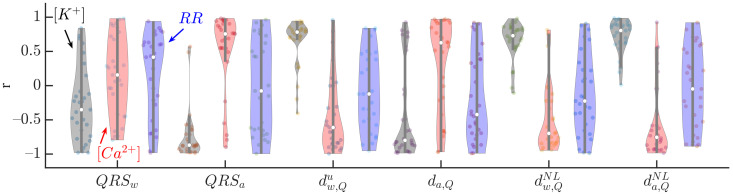
Pearson correlation coefficients between QRS markers ($QRS_{w}$, $QRS_{a}$, $d_{w,Q}^{u}$, $d_{a,Q}$, $d_{w,Q}^{\mathrm{NL}}$ and $d_{a,Q}^{\mathrm{NL}}$) and $\left[K^{+}\right]$ (black), $\left[{\mathrm{Ca}}^{2+}\right]$ (red) and RR (blue) for all patients at all HD points. The central white dot indicates the median. Each dot corresponds to an individual patient.

**Figure 6 sensors-22-02951-f006:**
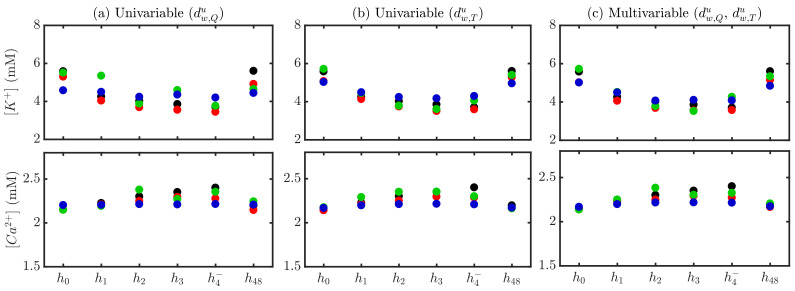
Actual (black) and estimated $\left[K^{+}\right]$ and $\left[{\mathrm{Ca}}^{2+}\right]$ for a patient using stage-specific (red), patient-specific (green) and global (blue) approaches. Univariable $d_{w,Q}^{u}$-based estimation is shown in (panel **a**), $d_{w,T}^{u}$-based in (panel **b**) and multivariable $d_{w,Q}^{u}$- $d_{w,T}^{u}$-based in (panel **c**).

**Figure 7 sensors-22-02951-f007:**
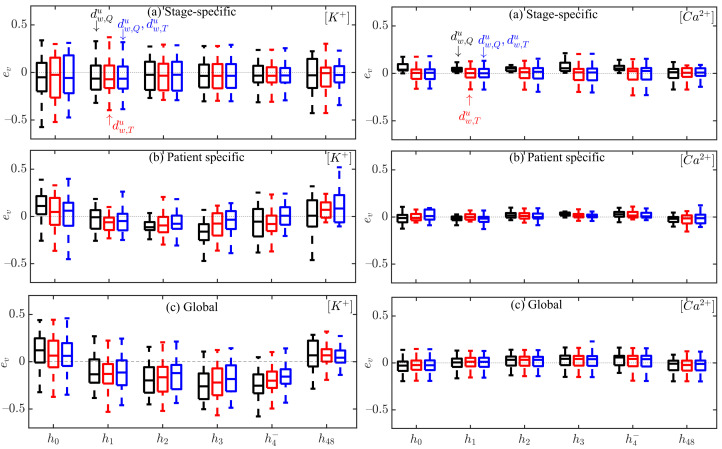
Box plots of $\left[K^{+}\right]$ and $\left[{\mathrm{Ca}}^{2+}\right]$ estimation errors $e_{v}$ during HD stages for all patients using $d_{w,Q}^{u}$(black), $d_{w,T}^{u}$(red) and the combination of $d_{w,Q}^{u}$ and $d_{w,T}^{u}$ (blue) for stage-specific (**top**), patient-specific (**middle**) and global (**bottom**) approaches. The central line indicates the median, whereas top and bottom edges show the 25th and 75th percentiles.

**Figure 8 sensors-22-02951-f008:**
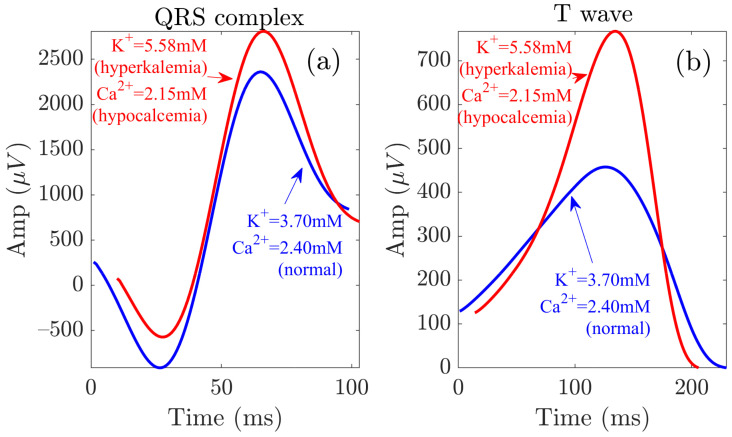
QRS and T wave variations at the start (red) and end (blue) of the HD session. Panels (**a**,**b**) show the waveforms related to the QRS complex and the T wave, respectively.

**Table 1 sensors-22-02951-t001:** Characteristics of the study population. Values are expressed as number (%) for categorical variables and median (interquartile range, IQR) for continuous variables. Reproduced with permission from Bukhari et al., Computers in Biology and Medicine; published by Elsevier, 2022.

Characteristics	Quantity
Age [years]	75(12)
Gender [male/female]	20 (69%)/9 (31%)
**Electrolyte concentrations**	
$\left[K^{+}\right]$ [Pre HD] (mM)	5.05(1.57)
$\left[K^{+}\right]$ [End HD] (mM)	3.35(0.62)
$\left[{\mathrm{Ca}}^{2+}\right]$ [Pre HD] (mM)	2.15(0.18)
$\left[{\mathrm{Ca}}^{2+}\right]$ [End HD] (mM)	2.32(0.2)
	#**Patients (%)**
**HD session duration**	
240 min	26(90%)
210 min	3(10%)
**Dialysate composition**	
Potassium (1.5 mM)	21(73%)
Potassium (3 mM)	5(17%)
Potassium (variable mM)	3(10%)
Calcium (0.75 mM)	8(28%)
Calcium (0.63 mM)	21(72%)

**Table 2 sensors-22-02951-t002:** *p*-values from the parametric test (*t*-test) to evaluate statistical significance of non-zero mean Fisher z-transformed Pearson correlation coefficients between QRS markers and $\left[K^{+}\right]$, $\left[{\mathrm{Ca}}^{2+}\right]$ and RR.

*p*-Values	${\mathrm{QRS}}_{w}$	${\mathrm{QRS}}_{a}$	$d_{w,Q}^{u}$	$d_{a,Q}$	$d_{w,Q}^{\mathrm{NL}}$	$d_{a,Q}^{\mathrm{NL}}$
$\left[K^{+}\right]$	0.01	<0.01	<0.01	<0.01	<0.01	<0.01
$\left[{\mathrm{Ca}}^{2+}\right]$	0.09	<0.01	<0.01	0.02	<0.01	<0.01
RR	0.37	0.94	0.48	0.12	0.23	0.60

**Table 3 sensors-22-02951-t003:** Actual and estimated $\left[K^{+}\right]$ and $\left[{\mathrm{Ca}}^{2+}\right]$ values over the study population at each HD stage using multivariable (m) estimation and stage-specific (S), patient-specific (P) and global (G) approaches. Values are expressed as median (IQR) and the units are mM.

Actual vs. Estimated $\left[K^{+}\right]$	$h_{0}$	$h_{1}$	$h_{2}$	$h_{3}$	$h_{4}^{-}$	$h_{48}$
$\left[K^{+}\right]$	5.10 (1.30)	3.90 (0.86)	3.64 (0.81)	3.40 (0.71)	3.40 (0.56)	5.08 (1.53)
$\left[{\hat{K}}^{+}\right]{}_{m}^{S}$	5.31 (0.43)	4.03 (0.18)	3.70 (0.08)	3.49 (0.09)	3.43 (0.05)	4.56 (1.21)
$\left[{\hat{K}}^{+}\right]{}_{m}^{P}$	4.76 (1.90)	4.01 (1.33)	3.84 (1.16)	3.46 (0.97)	3.28 (0.44)	4.43 (1.46)
$\left[{\hat{K}}^{+}\right]{}_{m}^{G}$	4.50 (0.71)	4.33 (0.40)	4.07 (0.33)	3.97 (0.26)	3.84 (0.19)	4.57 (0.75)
$\left[{\mathrm{Ca}}^{2+}\right]$	2.15 (0.20)	2.23 (0.20)	2.29 (0.19)	2.31 (0.23)	2.36 (0.21)	2.17 (0.20)
$\left[\hat{C}a^{2+}\right]{}_{m}^{S}$	2.13 (0.02)	2.21 (0.05)	2.25 (0.02)	2.31 (0.05)	2.28 (0.03)	2.07 (0.13)
$\left[\hat{C}a^{2+}\right]{}_{m}^{P}$	2.06 (0.29)	2.27 (0.23)	2.21 (0.26)	2.29 (0.20)	2.25 (0.23)	2.18 (0.20)
$\left[\hat{C}a^{2+}\right]{}_{m}^{G}$	2.19 (0.04)	2.20 (0.02)	2.22 (0.02)	2.23 (0.01)	2.23 (0.01)	2.19 (0.04)

**Table 4 sensors-22-02951-t004:** Intra-patient Pearson correlation coefficient *r* between actual and estimated $\left[K^{+}\right]$ using univariable and multivariable estimators, with stage-specific (S), patient-specific (P) and global (G) approaches. Values are expressed as median (IQR).

$r_{\left[K^{+}\right],\left[{\hat{K}}^{+}\right]}$	$d_{w,Q}^{u}$	$d_{w,T}^{u}$	$d_{w,Q}^{u}$, $d_{w,T}^{u}$
S	0.98 (0.08)	0.96 (0.06)	0.93 (0.30)
P	0.56 (0.75)	0.55 (0.90)	0.75 (0.51)
G	0.75 (0.15)	0.82 (0.35)	0.86 (0.32)

**Table 5 sensors-22-02951-t005:** Intra-patient Pearson correlation coefficient *r* between actual and estimated $\left[{\mathrm{Ca}}^{2+}\right]$ using univariable and multivariable estimators, with stage-specific (S), patient-specific (P) and global (G) approaches. Values are expressed as median (IQR).

$r_{\left[{\mathrm{Ca}}^{2+}\right],\left[\hat{C}a^{2+}\right]}$	$d_{w,Q}^{u}$	$d_{w,T}^{u}$	$d_{w,Q}^{u}$, $d_{w,T}^{u}$
S	0.88 (0.38)	0.88 (0.22)	0.80 (0.78)
P	0.88 (0.22)	0.63 (0.59)	0.63 (0.37)
G	0.64 (0.73)	0.64 (0.49)	0.70 (0.55)

**Table 6 sensors-22-02951-t006:** Estimation errors (*e*) using stage-specific (S), patient-specific (P) and global (G) approach-based $\left[K^{+}\right]$ estimators, from all patients at all HD time points. Values are expressed as mean ± standard deviation and the units are mM.

*e*	S	P	G
$d_{w,Q}^{u}$	−0.041±0.831	−0.091±1.419	−0.204±0.971
$d_{w,T}^{u}$	0.004±0.806	−0.147±0.809	−0.169±0.959
$T_{S/A}$	0.005±0.792	−0.157±1.120	−0.213±0.996
$T_{S/\sqrt{A}}$	0.003±0.811	−0.149±1.422	−0.238±1.048
$d_{w,Q}^{u}$ and $d_{w,T}^{u}$	0.073±0.808	−0.035±1.113	−0.144±0.883

**Table 7 sensors-22-02951-t007:** Estimation errors (*e*) using stage-specific (S), patient-specific (P) and global (G) approach-based $\left[{\mathrm{Ca}}^{2+}\right]$ estimators, from all patients at all HD time points. Values are expressed as mean ± standard deviation and the units are mM.

*e*	S	P	G
$d_{w,Q}^{u}$	0.117±0.134	−0.007±0.300	0.018±0.175
$d_{w,T}^{u}$	0.0003±0.170	0.024±0.178	0.018±0.172
$T_{S/A}$	−0.002±0.179	0.025±0.191	0.023±0.180
$T_{S/\sqrt{A}}$	−0.002±0.183	0.027±0.201	0.020±0.183
$d_{w,Q}^{u}$ and $d_{w,T}^{u}$	0.023±0.180	0.010±0.125	0.016±0.174

## Data Availability

The dataset is still ongoing and it is available upon request to the corresponding author.

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
