# Peer review of "Monitoring of Serum Potassium and Calcium Levels in End-Stage Renal Disease Patients by ECG Depolarization Morphology Analysis"

_sensors, 2022, doi:10.3390/s22082951_

Round 1
Reviewer 1 Report
This paper proposed QRS morphology markers for estimation of serum potassium and calcium in ESRD patients, and combined with T wave morphology markers to product the univariable and multivariable estimators to achieve a great prediction.
1. The paper has some writing errors. Please proofread it carefully.
For example: Page 4, the title of Figure 2, “performed in this study, . from…”
Page 5, when introducing Equation (6), miswrite it as “ can be quantified as”
2. In introduction section, line 53 on page 2, it’s written that “None of these studies have, however, investigated overall variations in the morphology of the QRS complex during HD”. But one of studies described above has assessed the sine wave shape of the QRS complex, please check if this statement is accurate.
3. What’s the meaning of the corner marker “m” in Table 3?
4. Why define another time point for ECG analysis? Is this a particular time point?
5. When computing MWQRS, are all12 lead ECGs used? How to compute initial MWQRS by aligning the QRS complexes with respect to their gravity center? Please describe the method and procedure in detail.
6. In the result section, line 233 on page 9, it’s written that “On the other hand, no marker presented significant association with RR”. However, studies have shown that RR interval is correlated with heart rate, for example [1]. Please explain how the QRS width is not significantly related to RR based on the data in this paper.
[1] Katerina Hnatkova, Peter Smetana, Ondrej Toman, Georg Schmidt, Marek Malik, Sex and race differences in QRS duration, EP Europace, Vol. 18, no. 12, pp. 1842–1849, 2016.
7. Were there any other diseases in the study population that could affect serum potassium and calcium concentrations? Does the duration of renal disease affect the levels of serum potassium and calcium?
8. QRS in ECG corresponds to phase 0 depolarization and phase 1 repolarization and its spreading process of ventricle, in which sodium also plays an important role, especially in phase 0. Is it reasonable to use QRS morphology to estimate serum potassium and calcium without considering change in sodium?
Reviewer 2 Report
The manuscript proposes using new markers from the morphology of the QRS complex of the ECG data. The ECG-based estimators of [K+] and [Ca2+] are derived from the QRS amplitude, duration and morphology. Computational results demonstrates that the ECG-based estimators derived from QRS can accurately predict [K+] and [Ca2+] with relatively small prediction error. Overall, the topic is worthy of affirmation and the results are promising. This is a well-written paper and can be considered for publication after a moderate revision.
1, In the introduction, it should discuss more about the recent publications which use other estimators for [K+] and [Ca2+].
2, The font size for Fig.8 axis labels are tool large compared with the other figures. Please revise.
3, In Eq. (11), the authors listed computing coefficient vector by Pseudo inverse. What is the computational cost using the proposed approach?
4, In Eq. (14), the relative errors were computed for performance measurement. How about other metrices such as Mean Absolute Error and Root-Mean-Square-Error? Please discuss
4, The conclusion section should be further improved. The authors should discuss what is the future research direction.
5, This is an interesting approach. The authors should cite the following papers and discuss how the proposed estimators can be transferred to the Energy field:
Li, H., Deng, J., Feng, P., Pu, C., Arachchige, D. D., & Cheng, Q. (2021). Short-Term Nacelle Orientation Forecasting Using Bilinear Transformation and ICEEMDAN Framework. Frontiers in Energy Research, 697.
Li, H., Deng, J., Yuan, S., Feng, P., & Arachchige, D. D. (2021). Monitoring and Identifying Wind Turbine Generator Bearing Faults Using Deep Belief Network and EWMA Control Charts. Frontiers in Energy Research, 770.
After revision, this manuscript has the strong potential for publication.
Reviewer 3 Report
Authors addressed a a very challenging clinical issue. Non invasive monitoring of serum electrolytes could be very useful for physicians and life saving for ESRD patients. Nevertheless there are still some points that should be further addressed. Why authors choose to analyze such a small sample size? Do they plan to perform an external cohort validation study of their findings? Is the majority of Holter devices available suitable for such recordings and analyses? Could all these calculations be prformed using a single standard 12leads ECG at any time point? Do the authors believe that such calculations could be applicable using 1lead ECG such that available by wearables? Baseline and serial electrolytes measurements were near normal. Do the authors feel confident that their algorithms will remain reliable for values outside the normal range? Could the any of these information acquired from QRS morphology analysis aid to ameliorate arrhythmia risk prediction? Do the combination of such a characteristic with the expected electrolyte value increase the accuracy for tisk prediction?Author Response
Please see the attachment.

Round 2
Reviewer 1 Report
Thanks to the authors for their detailed answers and revisions to the questions and suggestions raised. In response to your reply, there are still two doubts remained.
- As described in the authors' response, patients appeared a wider or narrower QRS or no change at high RR interval. Authors also observed a non-liner relationship between QRS width and RR interval. However, Table 2 in this paper presents no correlation between QRS width and RR interval. What causes the experimental data to appear such results?
- Table 3 in the paper shows that the median of [Ca2+] value does not vary by more than 0.5 mM at different HD stages, whereas Fig. S13 in Supplementary Material shows the median of [Na+] value beyond this range at different HD stages. This may not be strong evidence that sodium is much less remarkable than calcium and does this reflect a significant change in sodium along HD? In addition, the cardiac Na+channel (hNav5) is primarily responsible for the depolarization of atrial and ventricular myocytes. Blockade of the Na+ current may manifest as prolongation of the QRS complex on the ECG which is the primary mechanism of some drugs for cardiac arrhythmias. Is it reasonable to state no QRS markers was strongly associated with [Na+] only based on 29 ESRD patients? Moreover, Pearson correlation coefficient is a linear correlation coefficient, which has limitations in reflecting the relationship between [Na+] and QRS markers.
Reviewer 2 Report
Authors revised the manuscript and the current version can be considered for publication.
Author Response
We are grateful to the Reviewer for the revision and for the helpful and constructive comments. It is our belief that the manuscript has been substantially improved after making the suggested modifications and additions. Thank you so much!
Reviewer 3 Report
Thank you very much for the time you spent to answer the comments and to revise your original manuscript. I am still a bit hesitant regarding the clinical utility of your methods, but not only this is an interesting concept but you have also done a great job. I am looking forward to see if such methods could alter our everyday practice, not only in ESRD patients.
Author Response
We appreciate the careful review and constructive suggestions made by the Reviewer. It is our belief that the manuscript has been substantially improved after making the suggested modifications and additions. We believe that combination of depolarization and repolarization markers to estimate serum electrolytes could improve the accuracy for risk prediction and to improve identification of arrhythmic risk.
Round 3
Reviewer 1 Report
Thanks to the authors for revising again based on our questions and suggestions. A specific analysis of the relationship between QRS width and RR interval in the results has been added to the latest version of the paper. Meanwhile, the effect of [Na+] on QRS complex in 29 ESRD patients was fully analyzed. It is expected that the effect of [Na+] on QRS complex will be fully considered when using QRS markers in further study. The current version of the paper is qualified for publication.
Author Response
We would like to thank you for the opportunity to revise and improve our manuscript. We appreciate the careful review and constructive suggestions made by the Reviewer. It is our belief that the manuscript has been substantially improved after making the suggested modifications and additions.